# Analysis of Korean Fencing Club Members’ Participation Intention Using the TPB Model

**DOI:** 10.3390/ijerph18062813

**Published:** 2021-03-10

**Authors:** Young-Jae Kim, E-Sack Kim

**Affiliations:** Department of Physical Education, Chung-Ang University, Seoul 06974, Korea; yjkim@cau.ac.kr

**Keywords:** fencing, fencing club members, Theory of Planned Behavior, participation intention, Korea

## Abstract

This study aims to investigate effects of three factors from the Theory of Planned Behavior (TPB)—attitude, subjective norms, and perceived behavioral control—on 233 fencing club members’ intention to continue participation, while considering fencing’s low popularity as a sport in Korea. This study analyzed data from members of fencing clubs in Seoul, Gyeonggi, Daegu, and Busan, using frequency analysis, exploratory factor analysis, reliability analysis, correlational analysis, and multiple regression with SPSS Windows 25.0 software. Results indicate that selected TPB model factors—specifically attitude and subjective norms—positively affected intention to continue participating. Moreover, significant influences of attitude and subjective norms were found in both men and women. Attitude significantly influenced intention in participants in their twenties, thirties, and forties or over; subjective norms significantly influenced intention in participants in their twenties and forties or over; and perceived behavioral control significantly influenced intention in participants in their thirties. Lastly, attitude and subjective norms significantly influenced intention when subjects participated once a week or twice a week and at least three times a week, and perceived behavioral control significantly influenced intention only when they participated at least three times a week. This suggests that members perceived their participation in fencing favorably and that the people around them encouraged them to continue participation in fencing and viewed it as a positive activity. The findings may be useful for understanding how to further popularize fencing in Korea and encourage current club members to maintain or increase their participation levels.

## 1. Introduction

The development of sports in South Korea (henceforth, Korea) was stimulated by the enactment of the National Sports Promotion Act in the 1960s, and as a result of this enactment, sports began to be further promoted; subsequently, the country hosted the 1986 Asian Games in Seoul and the 1988 Olympics in Seoul [1]. In particular, the remarkable performances of individual elite athletes such as Yuna Kim (Figure skating) and Taehwan Park (Swimming) have greatly advanced the level of Korean sports participation, and outstanding performances in team sports such as baseball and soccer have also attracted nationwide interest and support [2].

Based on such exceptional athletic performances, Korea is consistently ranked among the top 10 countries in international sports competitions, ranking 5th in the 2010 Winter Olympics in Vancouver, 2nd in the 2014 Asian Games in Incheon, and 5th again in the 2020 Olympics in London. Additionally, it became the fourth country to host all five mega sporting events, including the 1988 Seoul Olympics, the 2002 Korea–Japan World Cup, the IAAF World Championships in Daegu in 2011, the 2018 Winter Olympics–PyeongChang, and the 2019 World Aquatics Championships in Gwangju [3].

Despite the country’s astounding advances and growth, there is a serious discrepancy that needs to be resolved that is marked by the independent and mutually exclusive relationships among school sports, professional sports, and recreational sports, which has fueled growth in a distorted and unfavorable direction [4].

In particular, challenges in the promotion of unpopular sports are especially concerning. Although the distinction between popular sports and unpopular sports is vague, popular sports can be thought of as those that have gained recognition as professional sports that enjoy rich infrastructures and private club activities. However, other less well-known sports are often considered unpopular [5]. The majority of unpopular sports have low levels of governmental support and management, receive no attention from the public, and have difficulty attracting athletes [2].

Although studies for invigorating unpopular sports have found a variety of results, these studies are about policy measures or sponsorship and organization management, which are also far from realistic plans [6]. According to Han et al. [2], to invigorate unpopular sports, there is a need to get title sponsorship, use social networking services and mobile advertisements, and benchmark successful examples of other sports, or make active use of diverse media.

Fencing is an example of an unpopular sport in Korea [7]. Fencing has garnered more participants in recent years, and since the 2012 London Olympics, the number of fencing clubs in Korea has risen from 3 or 4 to 78, with a total of 2676 members [8], which is low compared to other popular lifetime sports such as football (4054 clubs) and badminton (5852 clubs) [9]. In Korea, fencing is becoming an event in which athletes reliably win many medals because of Korea’s excellent records in various international competitions, including the Olympics [10].

In contrast to football and basketball, fencing is not an easily accessible sport even in the United States, which is the global leader in recreational sports. It requires expensive equipment, and participating in domestic and international competitions is extremely costly. For this reason, fencing is usually enjoyed by wealthy Caucasians as opposed to Africans, South Americans, and Asians [11]. Furthermore, it was revealed that some people used fencing to enter Harvard University, which is one of the Ivy League schools [12]. Similarly, rich South Koreans use fencing as an entry point to top ranking universities [13].

Jeon [14] reported that Korean fencing clubs are receiving more than 100 new members every year. However, simply examining the number of new participants cannot shed light on the reasons for the increased popularity of fencing. Therefore, it is important to examine these members’ participation intention, especially since fencing is an unpopular sport and it faces various obstacles to participation and consequent continuation behaviors.

If growth in the sport of fencing is to continue in Korea, it is important to understand participation intention among members of recreational fencing clubs. Kim [15] reported that understanding participation intention can illuminate the reasons underlying people’s participation in sports activities. The strength of the intention underlying a specific behavior determines the likelihood of increased engagement in that specific behavior—that is, stronger intention leads to increased behavior occurrence. Plotnikoff et al. [16] found that 35% of changes in exercise behavior can be explained by intention and perceived behavioral control (PBC), and 40% of changes in intention can be predicted by subjective norms and PBC.

The Theory of Planned Behavior (TPB) by Ajzen [17,18] can contribute to explaining participation intention and help in examining why Korean fencing club members choose to participate in fencing despite its unpopularity. Ajzen and Fishbein [19] further developed the TPB model by expanding the Theory of Reasoned Action (TRA) to include PBC [17,18,20]. TPB holds that the intention underlying a certain behavior is a direct determinant of that particular behavior’s occurrence; furthermore, participants’ intention is determined by attitude, subjective norms, and PBC [17,18]. TPB-related social and health behaviors have been studied, and intention was shown to be a major predictor of actual behaviors [21]. According to Hagger, Chatzisarantis, and Biddle [22], TPB is defined by behavioral intention and PBC. Attitude indicates a positive or negative assessment of behavioral performance, subjective norms refer to pressure by social expectations and surrounding others on behavioral performance, and PBC represents the constraints that time, money, and opportunity impose on actual behavior. Research has shown that PBC has both indirect and direct influences on intention [23]. Furthermore, attitude, subjective norms, and PBC showed an explanatory power of 45% in relation to intention [22].

Courneya [24] argued that the TPB is a useful model for understanding why people engage in exercise. Hagger and Chatzisarantis [25] showed that TPB is an appropriate theory for predicting participation intention with regard to sports activities and actual behaviors and that attitude and PBC were the most powerful predictors of physical activity participation behavior. Likewise, research by Downs and Hausenblas [26] reported that attitude and PBC showed a high predictive power in the TPB model.

Several previous studies have empirically examined adults’ participation in sports or physical activities using the TPB [27,28,29,30,31]. However, even though increasing numbers of Korean people are participating in unpopular sports and various types of public media are fueling continued interest in such sports, few or no studies have used TPB to analyze participation intention among participants of unpopular sports clubs, especially fencing. Understanding the psychology underlying participants’ participation behaviors could help uncover the reason for their participation in fencing despite its unpopularity. Therefore, using TPB to identify social and psychological participation-related factors among fencing club members could provide foundational data for promoting the balanced development of various unpopular recreational sports, including fencing.

According to a study by Rhodes, Blanchard, and Blacklock [32], personal intention to do a physical activity does not vary depending on sex. However, a study by Plotnikoff et al. [16] noted that PBC had a high predictive power of behavior depending on gender, while attitude had the highest correlation with intention.

Nigg, Lippke, and Maddock’s [33] research indicates that there were no gender differences with regard to attitudes, subjective norms, or PBC when engaging in physical activities. They add that the younger the age, the lower the correlations between intention and behavior. Furthermore, Kim [34] defined people who currently exercise for at least 30 min a week and less than 3 weeks as “preparation participants” and those who have been exercising for over 6 months as “maintenance participants.” The American College of Sports Medicine [35] classifies people who participate in sports activities for over 30 min at a time, 3 times a week or more, as regular exercisers.

Hence, this research attempts to investigate the associations of fencing club members’ attitudes, subjective norms, and PBC on intention. To accomplish the aims of this research, the following hypotheses were proposed.

**Hypothesis** **1** **(H1).**
*Attitude, subjective norms, and PBC regarding fencing club members’ fencing club participation will positively influence their intention.*


**Hypothesis** **2** **(H2).**
*Attitude, subjective norms, and PBC regarding club participation will positively influence intention depending on the gender of fencing club members.*


**Hypothesis** **3** **(H3).**
*Attitude, subjective norms, and PBC regarding fencing club participation will positively influence intention depending on the age of fencing club members.*


**Hypothesis** **4** **(H4).**
*Attitude, subjective norms, and PBC regarding club participation will positively influence intention depending on fencing club members’ weekly participation rate.*


## 2. Materials and Methods

### 2.1. Participants

This study was conducted among fencing club members. The questionnaire was administered to members of fencing clubs in Seoul, Gyeonggi, Incheon, Daegu, and Busan from January 2019 to June 2019, and the researchers contacted each fencing club to request permission to visit and distribute the questionnaire. The researchers visited fencing clubs in person to distribute research consent forms to the subjects and collected the signed forms, which indicated the subjects’ agreement to participate in the study. Our questionnaires, which were based on a self-report method, were distributed through convenience sampling, a type of non-probability sampling. A total of 300 questionnaires were distributed, 261 of which were retrieved. After excluding 28 questionnaires with careless or missing responses, 233 questionnaires were used as data for the study.

### 2.2. Measurements

The survey questions examined four demographic characteristics: sex, age, occupation, and monthly income. The degree of fencing activity involvement was assessed based on five items: weekly participation frequency, the duration of each fencing session, current fencing performance, weekly fencing lesson attendance frequency, and length of participation. The TPB contains 22 items; this study utilized seven items for attitude, four items for subjective norms, four items for perceived behavioral control, and seven items for intention to continue behavior. Additionally, the questionnaire used in this study was translated into Korean and validated as a measure of TPB factors among Korean people [36]. Content validity of the questionnaire was analyzed by two experts (one sports sociology professor and one expert with a Ph.D. in sports sociology).

### 2.3. Theory of Planned Behavior

TPB was measured using a modified version of the questionnaire developed by Ajzen [17] and applied to a study by Kim [36]. The questionnaire contained 22 items: seven items for attitude, four items for subjective norms, four items for perceived behavioral control, and seven items for intention to continue a given behavior; each item was rated on a 5-point Likert scale from “strongly disagree” (1) to “strongly agree” (5). Confirmatory factor analysis (CFA) and exploratory factor analysis (EFA) were performed to validate the study’s measure of TPB. In the study by Kim [36], the Cronbach’s α coefficients of the TPB model for intention, attitude, subjective norms, and PBC were 0.87, 0.72, 0.79, and 0.74, respectively, which means that the TPB model has internal consistency reliability. The study by Kim [36] demonstrated that, overall, the CFA results (χ^2^ = 233.39, df = 111, *p* < 0.01, goodness-of-fit index = 0.94, comparative fit index (CFI) = 0.93, non-normed fit index = 0.91, root-mean-square residual (RMR) = 0.054) had appropriate model indexes for the TPB measure.

The reliability of the data was measured using Cronbach’s α; the relevant values were 0.95 for intention to continue behavior, 0.90 for attitude, 0.87 for subjective norms, and 0.71 for perceived behavioral control, which therefore confirmed the reliability. Factor analysis resulted in four factors, and to establish the validity of each factor, a factor loading cutoff of 0.5 or higher was used; based on this, two factors were excluded (subjective norms (SN) Q4: I do fencing because I was recommended by my friends or family; PBC Q3: I can afford to do fencing club activities).

When Korean people’s intention to participate in leisure sports was analyzed using the TPB model in the study by Song and Park [37], normed fit index (NFI), CFI, and root-mean-square error of approximation (RMSEA) values of CFA were 0.94, 0.95, and 0.08, respectively, which means the TPB model was appropriate.

To determine the factor structure in this study, CFA was conducted. As a result, permissible fitness values were obtained: χ^2^ = 470.289, df = 164, *p*  < 0.00, NFI = 0.87, CFI = 0.91, incremental fit index = 0.91, RMSEA = 0.09.

### 2.4. Procedure and Data Analysis

Study data were analyzed using SPSS 25.0 software (IBM, Armonk, NY, USA) for Windows. First, the demographic data were analyzed using descriptive statistics—namely, frequency analysis. Second, based on the study by West et al. [38], skewness in the TPB model ranged from −1.012 to −0.511 and kurtosis ranged from −1.900 to −0.107, which means there was a normal distribution. Third, construct validity testing was performed with CFA and EFA to examine the factor structure, and reliability testing was performed using Cronbach’s α to establish the reliability of the tool. Fourth, *t*-tests, and one-way ANOVAs were performed to compare the differences in the TPB model according to demographic factors. Lastly, associations among the variables were analyzed with correlation analysis. For converting sex (male = 1, others = 0), age (participant in his or her forties = 1, others = 0), and the number of weekly sessions (at least three times a week = 1, others = 0), which are categorial variables among demographic variables that showed significant differences from dependent variables, into continuous variables, multiple regression analysis using dummy variables was conducted.

## 3. Results

EFA was performed to determine tool validity and population normality. Bartlett’s test of sphericity result was 3500.045, and the Kaiser–Meyer–Olkin value was 0.917 (*p* < 0.000), with a cumulative variance of 71.766. Table 1 shows the results of EFA for TPB.

Table 2 shows the demographic data and descriptive statistical values for the fencing club members. First, the male participants (*n* = 149, 63.9%) outnumbered the female participants (*n* = 84, 36.1%), and in terms of age, people in their 20s showed the highest participation rates (*n* = 132, 56.7%). Most of the participants were students (*n* = 92, 39.5%), and the most common monthly income level for the participants was <1.5 million KRW (*n* = 92, 39.5%).

Table 3 shows the results of the correlation analysis, which was applied to the factors to determine discriminant validity. The TPB factors were positively correlated to each other (correlation coefficient was lower than 0.800), thereby satisfying the criteria for multicollinearity; this indicated that there were no problems in the analysis [39].

Table 4 shows the results of a multiple regression analysis that was conducted to identify the predictor factors for intention to continue participation among fencing club members. The results showed that the TPB factors explained 40% of the variance, and attitude (β = 0.38) had the highest significant effect on intention, followed by subjective norms (β = 0.27).

Table 5 shows the results of multiple regression that was conducted to examine the significance of the effects of TPB factors on intention with regard to participants’ demographics. After control variables were applied, the explanatory power predicting the TPB factors in different sex groups was 41%, and attitude (β = 0.39) had the biggest influence on intention, followed by subjective norms (β = 0.28). Different age groups was 42%, and attitude (β = 0.36) had the biggest influence on intention, followed by subjective norms (β = 0.27). The number of weekly sessions was 40%, and attitude (β = 0.38) had the biggest influence on intention, followed by subjective norms (β = 0.27).

Among men, the factors explained 41% of the variance, and attitude (β = 0.40) had the strongest significant effect on intention, followed by subjective norms (β = 0.26). Among women, the factors explained 41% of the variance, and attitude (β = 0.35) had the strongest significant effect on intention, followed by subjective norms (β = 0.33). Among participants aged 20–29 years, the factors explained 35% of the variance, and subjective norms (β = 0.35) had the strongest significant effect on intention, followed by attitude (β = 0.29). Among participants aged 30–39 years, the factors explained 48% of the variance, and attitude (β = 0.55) had a significant effect on intention. Finally, among participants aged 40–49 years, the factors explained 47% of the variance, and subjective norms (β = 0.42) had the strongest significant effect on intention, followed by attitude (β = 0.34). In terms of weekly participation frequency, the factors explained 40% of the variance, and attitude (β = 0.43) had the strongest significant effect on intention, followed by subjective norms (β = 0.28) among those who participated in fencing 1–2 times a week. Among those who participated in fencing 3 or more times a week, the factors explained 43% of the variance, and subjective norms (β = 0.27) had the strongest significant effect on intention, followed by attitude (β = 0.27) and perceived behavioral control (β = 0.26).

## 4. Discussion

This study used the TPB model (attitude, subjective norms, and perceived behavioral control) to examine fencing participation intention among fencing club members. The study results are discussed below.

The participants showed a positive attitude toward fencing. When analyzing the effects of attitude, subjective norms and PBC regarding the participation of fencing club members in fencing clubs on intention, attitude and subjective norms showed significant effects, which means Hypothesis 1 was accepted. This suggests that they perceived it to be a good sport and an enjoyable positive activity. This also suggests that the likelihood of continuously participating in recreational fencing increases with increasingly positive attitude [37]. Furthermore, subjective norms had a positive effect on participation intention; thus, positive perceptions and attitudes toward fencing from people around them further promoted their participation in fencing clubs. Many previous studies on TPB have reported empirical support for the influence of attitude [22,40]. Likewise, this study demonstrates that attitude toward specific behavior positively influenced intention.

Lee and Suh [41] stated that positive feedback and communication increased exercise participation through social pressure from surrounding people. This is similar to the effect of high subjective norms on fencing club members. The findings of previous studies support those of this study by showing a significant relationship between subjective norms and behavioral intention [42,43]. Others’ support, which is one of the reasons fencing club members participated in fencing in the first place, encouraged not only their own participation but also the participation of people around them. Therefore, it may be necessary to launch diverse events to continue to engage people in fencing. Moreover, utilizing and managing positive external factors, exercise programs, facilities, programs, and advertisements could also help to motivate the people close to fencing club members, such as friends and families, to participate in the sport as well. In turn, this can further encourage fencing club members to continue their participation.

When analyzing the effects of attitude and subjective norms regarding the participation of fencing club members in fencing clubs on intention depending on gender, the variables showed significant effects, which means Hypothesis 2 was accepted. This shows that both men and women with more favorable attitudes toward fencing club participation were more likely to continuously participate in the clubs. Boiché and colleagues [44] showed that gender stereotypes could affect people’s perceptions and behaviors with regard to sports or dance participation. It has been suggested that men tend to prefer activities that require more physical movements and are strenuous and competitive [45], but in comparison, women prefer sports that require fewer physical movements and expose themselves to less competitions with others [46]. However, the present study showed that both men and women shared a highly positive attitude toward the same sport. Fencing is a competitive and dangerous sport that involves the use of weapons. Nevertheless, women participants perceived fencing to be a positive activity just as much as the men did. Jin [47] stated that the characteristic impressions of fencing—namely, gentlemanly behavior, sportsmanship, agility, and power—were the factors that motivated people to participate in the sport. In other words, these characteristics unique to fencing could help improve attitude by creating positive perceptions among participants. Thus, to promote fencing participation, it may be important to further highlight certain novel fencing-related experiences that are not provided by other types of sports. Moreover, promotions and marketing strategies that emphasize the positive aspects of fencing, irrespective of gender, would support both men’s and women’s satisfaction with fencing club participation.

When analyzing the effects of attitude and subjective norms regarding the participation of fencing club members in fencing clubs on intention depending on age, the variables showed significant effects, which means Hypothesis 3 was accepted. The analysis found that these factors had significant effects on intention among members in their twenties. This suggests that having positive expectations about exercise increases the likelihood of planning and deciding to participate in exercise; furthermore, sports participation is also influenced by others (e.g., friends, family, and so on) [48]. This result may be attributable to certain characteristics of people in their twenties (for example, the desire to pursue new challenges) [49].

In addition, attitude had significant effects on people in their thirties. The participants’ own attitude had a positive impact on their fencing club involvement. In contrast, Salmon and colleagues [50] reported that younger people who engaged in physical activities attracted more interest and were subject to more regulation from the people around them, thus suggesting that subjective norms had a weaker association with participation intention. In the present study, subjective norms had significant effects on such intentions among a younger age group (20–29 years); this showed that while people in their twenties preferred to spend time with other people, people in their thirties participated solely for their own satisfaction. Among people in their forties, attitude and subjective norms showed significant effects. Members of this population perceived fencing as a positive activity. It seems that the older people are, the more active they are when participating in sports, because they have sufficient health, financial resources, and time to engage in sports more actively [51]. According to a study by Beauchamp et al. [52], when people of a similar age participated in a physical activity together, their preference for the activity rose, and this led to an increase in the rate of participation. As suggested by Beauchamp [52], it seems that fencing club participants in their twenties and forties or over showed a high influence of subjective norms, because people of a similar age were joining their fencing clubs. It is, therefore, supposed that the more participants there were of a similar age, the higher the fencing club participation rate. Plans to encourage acquaintances with no interest in fencing to participate in fencing clubs [53] may further increase the number of fencing club participants since physical activities with others can boost and maintain participation, as revealed in Beauchamp’s study.

Morris, Venkatesh, and Ackerman [54] revealed differences in gender and age depending on the TPB sub-factors—namely, men were more influenced by attitude than women, and women were more influenced by subjective norms when they were older [32]. Nyman [55] stated that older people were more significantly influenced by PBC of physical activity participation, and, in particular, younger women had a lower level of PBC than did men. This demonstrates that attitude and subjective norms play a crucial role in intention. These results are consistent with our findings. As attitude had significant effects in this study, implementing age-specific feedback from instructors and ensuring the availability of professional expertise [56] could attract more participation in fencing.

When analyzing the effects of attitude, subjective norms, and PBC regarding the participation of fencing club members in fencing clubs on intention depending on number of weekly sessions, the analysis showed significant effects, which means Hypothesis 4 was accepted. Regarding the effects of attitude, subjective norms, and PBC on intention (based on weekly participation frequency), attitude and subjective norms had significant effects among members who participated 1–2 times a week; furthermore, attitude, subjective norms, and PBC all had significant effects on intention among members who participated 3 or more times a week. However, B value was lower than in the group with a lower ratio of participation. The study by Kim and Kim [57] implied that attitude and subjective norms could be high among people who first participate in physical activities, since it is fun, or recommendations or behaviors of others influence their participation, whereas among people who continue to participate in physical activities, attitude and subjective norms could be lower than those that first participate, as it is a part of their ordinary life. Lee, Kim, and Park [58] proposed a phased change model for individual behavioral changes, where people who exercised more than 3 times a week were referred to as regular exercise participants. They stated that people who exercised regularly had either experienced or were currently experiencing various physical or psychological benefits through exercise depending on their participation duration. This suggests that people who participate in fencing 3 or more times a week are participating more frequently because they have reaped positive benefits physically and emotionally from greater fencing involvement.

Myers and Roth [59] argued that exercise-related benefits promoted exercise participation behavior, while obstacles, including time, individual physical characteristics, and social and environmental factors, could hinder exercise participation. In this study, people who participated 3 or more times a week invested more time and enjoyed fencing more compared to those who participated 1–2 times a week. However, PBC had significant effects. In Korea, fencing is not an easily accessible sport, and there are certain spatial limitations on fencing club activities [60]. Furthermore, a study by Eyler et al. [61] reported that sports participation increased in inverse proportion to the sports facilities’ distance from participants’ homes. Babakus and Thompson [62] argued that not accurately understanding the positive physical and psychological advantages of physical activities or a lack of resources to participate in physical activities could decrease participation. Therefore, lowering mobility-related restrictions by, for example, promoting better geographical access and transportation access could increase participation in fencing. In addition, it is necessary to implement various marketing strategies to boost participants’ positive and favorable psychological attitudes toward fencing. Well-equipped facilities and diverse services could increase user satisfaction, which in turn could lead to continued participation.

To summarize the information presented above, the results of this study suggest that accentuating fencing’s image positively by organizing various events in fencing clubs, providing diverse programs, and offering opportunities to experience special events such as competitions, could make fencing more fun for club members. Additionally, promoting positive elements unique to fencing regardless of gender can increase participation of people from different age groups, and minimizing geographical distance and providing convenient transportation could lead to a higher ratio of participation. Ultimately, more is required to expand further promotion of fencing as a sport to the general public.

The following limitations are recognized. First, this study targeted people participating in fencing, which is an example of an unpopular sport. Therefore, further research needs to analyze more specific categories of leisure sports participants, such as those in seasonal sports, indoor and outdoor sports, racket sports, and ball sports, which would be useful in forecasting intention to participate in each sport. Moreover, controlling the social and psychological circumstances of diverse sports participants can be effective in reducing potential bias. Second, it was hard to control demographic properties used in this study and psychological or social circumstances perceived by fencing participants at the time of the survey. It is, therefore, considered that analyzing the levels of participation perceived by individuals through a mix of cross-sectional studies and longitudinal studies for predicting participation intentions in detail could help in seeking ways of increasing participation in fencing clubs. Third, some participants start involving themselves in fencing as a leisure sport because of external elements, such as public recognition of the sport as being aristocratic, its fancy movements, the use of French terminology, and so on. Thus, follow-up research should be conducted on external factors that motivate people to take up fencing, such as how people would like to present themselves to others. Fourth, this study conducted a quantitative study using a questionnaire. Unfortunately, there is a limit to in-depth analyses using a quantitative research method. Thus, an in-depth analysis through qualitative research is necessary to identify the reasons for participation in fencing.

## 5. Conclusions

This study investigated the effects of three select TPB model subfactors—attitude, subjective norms, and PBC—on fencing club members’ intention to continue participation in fencing, which has low popularity in Korea.

Multiple regression analysis revealed that attitude and subjective norms were related to participation intention among fencing club members. All fencing club members participated in fencing when they were able to access relevant facilities conveniently. Such participation could illuminate the differences between fencing and other combat sports. Due to the inherent dangers of combat sports, they tend to attract more male participation. However, this study showed that, regarding fencing, there were no gender gaps in terms of participation willingness and desire to participate along with close friends. In other words, both men and women perceived participating in fencing as a fun activity. Thus, to attract more people to participate in fencing, it may be necessary to consider various social characteristics, implement a variety of events that satisfy everyone’s needs regardless of gender, provide high-quality professional feedback by ensuring instructors’ expertise, and continuously show the positive aspects of fencing. In conclusion, fencing club members perceived participation in fencing favorably, and the people around them also encouraged their participation, which they further perceived positively. The findings may be useful for understanding how to further popularize fencing in Korea and for encouraging current club members to maintain or increase their participation levels.

## Figures and Tables

**Table 1 ijerph-18-02813-t001:** Results of exploratory factor analysis (EFA) for Theory of Planned Behavior (TPB) (*n* = 233).

Factors	Items	1	2	3	4	h^2^
Intention	Q4. I am willing to spend time and money to keep fencing in the future	0.86	0.16	0.19	0.07	0.94
	Q3. I am willing to keep fencing in the future	0.85	0.23	0.15	0.12	0.93
	Q2. I intend to keep fencing in the future	0.84	0.28	0.16	0.12	0.93
	Q7. I will practice fencing on a regular basis even after a year	0.83	0.21	0.11	0.05	0.94
	Q6. I will practice fencing on a regular basis even after six months	0.79	0.19	0.19	0.17	0.94
	Q1. I will try to keep fencing in the future	0.78	0.27	0.15	0.12	0.94
	Q5. I will practice fencing on a regular basis next month as well	0.74	0.27	0.24	0.19	0.94
Attitude	Q2. Fencing is a worthwhile activity	0.22	0.83	0.13	0.19	0.87
	Q3. Fencing is a beneficial activity	0.22	0.80	0.10	0.18	0.88
	Q1. Fencing is a positive activity	0.19	0.76	0.17	0.22	0.88
	Q7. Fencing is a meaningful activity for me	0.34	0.76	0.14	0.09	0.88
	Q6. Fencing is a dynamic activity for me	0.17	0.74	0.22	0.04	0.87
	Q4. Fencing is a necessary activity for me	0.39	0.70	0.08	0.09	0.88
	Q5. Fencing is a social activity for me	0.10	0.54	0.31	0.14	0.91
Subjective norms	Q1. My friends or family members think that it is good for me to practice fencing club activities	0.24	0.18	0.83	0.16	0.81
	Q2. My friends or family members support me in fencing club activities	0.28	0.24	0.81	0.17	0.76
	Q3. The people I value support my decision to join a fencing club	0.21	0.27	0.76	0.14	0.86
Perceived behavioral control	Q1. I can join a fencing club at any time I want	0.09	0.19	0.20	0.84	0.47
	Q2. Whether I join a fencing club or not depends on myself	0.10	0.39	0.01	0.75	0.61
	Q4. I have time to practice fencing club activities	0.30	0.05	0.26	0.61	0.74
Reliability		0.95	0.90	0.87	0.71	
Eigenvalue		5.377	4.527	2.474	1.975	
Eigenvalue (%)		26.884	22.633	12.371	9.877	
Cumulative variance (%)		26.884	49.517	61.888	71.766	

Note. 1 = Intention; 2 = Attitude; 3 = Subjective norms; 4 = Perceived behavioral control. Factor loadings below 0.50 have been removed.

**Table 2 ijerph-18-02813-t002:** Demographic characteristics of fencing club members (*n* = 233).

Characteristic	*n* (%)	AT	SN	PBC	INT
		Mean (SD)	Mean (SD)	Mean (SD)	Mean (SD)
Sex					
Male	149 (63.9)	4.41 (0.56)	4.19 (0.75)	4.20 (0.64)	4.41 (0.64)
Female	84 (36.1)	4.48 (0.45)	4.28 (0.69)	4.19 (0.63)	4.30 (0.54)
*p*-value		0.010	0.404	0.393	0.846
Age, M (SD)	28.01 (4.60)				
20–29	132 (56.7)	4.36 (0.56)	4.20 (0.74)	4.17 (0.65)	4.25 (0.63)
30–39	57 (24.5)	4.53 (0.45)	4.21 (0.77)	4.30 (0.66)	4.49 (0.54)
≥40	44 (18.9)	4.56 (0.49)	4.31 (0.63)	4.16 (0.61)	4.57 (0.55)
*p*-value		0.036	0.685	0.438	0.003
Occupation					
Student	92 (39.5)	4.37 (0.55)	4.29 (0.71)	4.25 (0.56)	4.29 (0.66)
Self-employed	24 (10.3)	4.40 (0.54)	4.01 (0.69)	4.05 (0.61)	4.40 (0.61)
Office worker	57 (24.5)	4.51 (0.48)	4.18 (0.73)	4.16 (0.69)	4.41 (0.60)
Service worker	14 (6.0)	4.46 (0.44)	4.19 (0.79)	4.17 (0.79)	4.44 (0.50)
Civic worker	20 (8.6)	4.51 (0.50)	4.28 (0.77)	4.30 (0.81)	4.47 (0.45)
Other	26 (11.2)	4.51 (0.59)	4.27 (0.78)	4.15 (0.77)	4.42 (0.59)
*p*-value		0.817	0.947	0.076	0.784
Monthly income					
<1.5 million KRW	92 (39.5)	4.39 (0.54)	4.33 (0.71)	4.28 (0.57)	4.29 (0.65)
1.5–2.49 million KRW	40 (17.2)	4.48 (0.49)	4.12 (0.66)	4.31 (0.61)	4.25 (0.56)
2.5–3.49 million KRW	53 (22.7)	4.39 (0.56)	4.03 (0.79)	4.00 (0.76)	4.37 (0.56)
≥3.5 million KRW	48 (20.6)	4.55 (0.49)	4.33 (0.70)	4.17 (0.61)	4.62 (0.54)
*p*-value		0.164	0.468	0.171	0.378
Weekly participation frequency					
1–2 times/week	143 (61.4)	4.49 (0.48)	4.24 (0.73)	4.13 (0.66)	4.36 (0.60)
≥3 times/week	90 (38.6)	4.36 (0.59)	4.20 (0.73)	4.29 (0.60)	4.40 (0.61)
*p*-value		0.005	0.641	0.163	0.672
Duration of single session					
<1.5 h	99 (42.5)	4.31 (0.52)	4.12 (0.81)	4.22 (0.67)	4.29 (0.59)
≥2 h	134 (57.5)	4.54 (0.51)	4.30 (0.65)	4.13 (0.62)	4.43 (0.61)
*p*-value		0.633	0.003	0.772	0.295
Length of participation					
<2 years	113 (48.5)	4.42 (0.51)	4.15 (0.73)	4.14 (0.63)	4.24 (0.57)
≥2 years	120 (51.5)	4.46 (0.54)	4.29 (0.72)	4.25 (0.65)	4.49 (0.62)
*p*-value		0.001	0.062	0.156	0.082

Note. AT: attitude, SN: subjective norms, PBC: perceived behavioral control, INT: intention.

**Table 3 ijerph-18-02813-t003:** Correlation analysis (*n* = 233).

Factor	Attitude	Subjective Norms	PBC	Intention
1. Attitude	1			
2. Subjective norms	0.523 ***	1		
3. PBC	0.492 ***	0.461 ***	1	
4. Intention	0.575 ***	0.523 ***	0.422 ***	1

*** *p* < 0.001.

**Table 4 ijerph-18-02813-t004:** Effects of TPB factors on intention (*n* = 233).

Dependent Variable	Independent Variable	Non-Standardized Coefficient		β	*p*-Value	*R* ^2^
		B	Standard Error			
Intention	Constant	1.05	0.28		0.000 ***	0.40
	Attitude	0.43	0.07	0.38	0.000 ***	
	Subjective norms	0.23	0.05	0.27	0.000 ***	
	PBC	0.10	0.06	0.11	0.073	

*** *p* < 0.001.

**Table 5 ijerph-18-02813-t005:** Effects of TPB factors on intention based on demographic characteristics of fencing club members (*n* = 233).

Selected Variable		Independent Variable	Non-Standardized Coefficient		β	*p*-Value	*R* ^2^
			B	Standard Error			
Control variable	Gender	Constant	0.91	0.28			0.41
		control	0.15	0.06	0.12	0.015 *	
		Attitude	0.44	0.07	0.39	0.000 ***	
		Subjective norms	0.23	0.05	0.28	0.000 ***	
		PBC	0.09	0.06	0.10	0.096	
	Age	Constant	10.06	0.274			0.42
		control	0.164	0.079	0.11	0.009 **	
		Attitude	0.415	0.073	0.36	0.000 ***	
		Subjective norms	0.226	0.051	0.27	0.000 ***	
		PBC	0.115	0.057	0.12	0.045 *	
	Number of weekly sessions	Constant	10.04	0.28		0.000 ***	0.40
		control	−0.03	0.06	−0.01	0.841	
		Attitude	0.44	0.07	0.38	0.000 ***	
		Subjective norms	0.23	0.05	0.27	0.000 ***	
		PBC	0.10	0.06	0.11	0.074	
Gender	Male	Constant	10.02	0.34		0.003 **	0.41
		Attitude	0.45	0.09	0.40	0.000 ***	
		Subjective norms	0.22	0.06	0.26	0.001 ***	
		PBC	0.11	0.07	0.11	0.145	
	Female	Constant	10.03	0.48		0.034 *	0.41
		Attitude	0.42	0.13	0.35	0.001 ***	
		Subjective norms	0.26	0.08	0.33	0.002 **	
		PBC	0.07	0.09	0.08	0.434	
Age	20s	Constant	10.31	0.37		0.001 ***	0.35
		Attitude	0.32	10.00	0.29	0.001 ***	
		Subjective norms	0.29	0.07	0.35	0.002 **	
		PBC	0.07	0.08	0.08	0.379	
	30s	Constant	0.57	0.53		0.290	0.48
		Attitude	0.66	0.14	0.55	0.000 ***	
		Subjective norms	0.04	0.08	0.06	0.585	
		PBC	0.17	0.09	0.21	0.065	
	40s or older	Constant	0.81	0.62		0.195	0.47
		Attitude	0.38	0.15	0.34	0.015	
		Subjective norms	0.36	0.11	0.42	0.002 **	
		PBC	0.11	0.11	0.12	0.347	
Number of weekly sessions	Once a week–twice a week	Constant	0.82	0.38		0.034	0.40
		Attitude	0.55	0.10	0.43	0.000 ***	
		Subjective norms	0.23	0.07	0.28	0.001 ***	
		PBC	0.02	0.07	0.02	0.762	
	At least three times a week	Constant	10.08	0.40		0.009 **	0.43
		Attitude	0.28	0.11	0.27	0.017 *	
		Subjective norms	0.23	0.11	0.23	0.006 **	
		PBC	0.26	0.08	0.26	0.016 *	

* *p* < 0.05, ** *p* < 0.01, *** *p* < 0.001.

## Data Availability

The data presented in this study are available on request from the corresponding author.

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
