# Peer review of "Analysis of Korean Fencing Club Members’ Participation Intention Using the TPB Model"

_ijerph, 2021, doi:10.3390/ijerph18062813_

Round 1
Reviewer 1 Report
Thank you for possibility of making a review of this article. I am convinced that Author/s feel the importance of this study. However, I would like to point to some issues that need to be explained more/corrected or abondoned.
Your study aims to investigate utility of TPB model in assessment of participation of adults in fencing. My first concern pertains to the following issues: why TPB model you used as a basis of your study (instead of others present in the literature)? That could be explained better in Introduction section. And why Authors focus on fencing? It is a very interesting idea, especially in Korea which is a country with traditions in sport promotion around the world. Author/s write that it is an example of unpopular sport, emphasis this fact in all major parts of the article. I think it would be better to avoid such arbitral assessment. Popularity of fencing, as other disciplines, is connected with many cultural factors: history of sport in particular country, geographical localisation, internal policies etc. We have to remember that reader of IJERPH are international, this is the first reason for why I would not emphasise so much on popularity. Another one is that I feel that Author/s do not have a strong rationale to claim that fencing is an unpopular sport. Author/s do not give any strong scientific source which points to fencing as an example of unpopular sport. Another example, even if we will exclude questionnaires with missing responses , still Author/s distributed 300 of them! It is preety much therefore again, it raises questions concerning unpopularity of fencing in Korea. Therefore I would revise the title of the article and focus just on fencing. In my opinion, it wouldn't diminish the overal value of this study in any way.
In Introduction I would like to read more about TPB model and detailed characteristics of the subcomponents: attitude (toward what and how?/in which manner?), subjective norms (what kind of norms? how these norms may manifest in decisions concerning decisions like "I will go on, I will participate" or "I quit") and perceived behavioral control (the same, needs to be explained better). This is a very important issue, because in Abstract Author/s write on lines 9-10 that "This study analyzed the attitudes of members of fencing clubs in Seoul, Gyeonggi, Daegu, and Busan". This sentence is misleading because we do not understand what attitudes Author's investigate. I would recommend to delete this sentence.
Author/s formulate 12 hypothesis. First of all, it is much too much. It is possible to reduce their number by joining some of them into on, e.g: Fencing club members' attitude, behavioral control and subjective norms forward participation in fencing club will positively influence intention. Instead 3 - we have 1. I also wonder if Author's have any assumptions concerning which component of TPB has the strongest influence on participation in sport like fencing. I think hypothesis should predict some outcomes. Now they don't predict anything. It is necessary to specify them more. Why Author/s didn't use the original version of TPB questionnaire?
Lines 99-100 - research in which using sunblocks by men and women was analysed doesn't make sense for me. Instead of this research, Author/s should search for data on psychological factors other than sex and age wich are connected strictly with the main topic (with reference to TPB and ToRA models).
Discussion and Conclusions - should be revised and pertain to the issues raised in Introduction (research, the main aim of this study and results). Author/s should discuss results more, as people from different phases of adulthood were examined. I think they should discuss the results with reference to personality traits, different social roles, experiences etc.
Because components of TPB were not characterized, it is difficult to comprehend the overall line of argument in this part of the text. For example, attitude was more likely to have significant effects among men than women.
I wish this review was more favorable, but I believe Author/s will revise their work and resubmit the manuscript.
Author Response
Dear reviewer,
We are very grateful for your comments about the manuscript. According to your previous advice, we amended the relevant parts of the paper. All revisions to the document have been clearly highlighted in the paper. After these revisions (your professional comments), the quality of this article has been greatly improved. I attached the specific modifications as a file.
Thank you very much for reviewing your thesis.
Kind regards,

Reviewer 2 Report
Thank you for offering such an interesting work. I was pleased to see the application of behavioral theory in the realm of sport. It is obvious you have worked to produce this manuscript. That should be applauded. I do believe that the document should be edited for language and grammar due to consistent and pervasive errors. This is an issue the can be remedied quickly. I have offered general comments regarding the respective portions of the manuscript below. I hope you find them helpful.
Abstract -
Please include population information and statistical information for the respective results mentioned.
Introduction -
Is there a need to specify you are speaking of South Korea (I apologize if that equation is politically insensitive)?
Some information seems repeated.
I am not sure the entire list of hypotheses are needed in the Introduction, perhaps a general statement regarding the aim of the research?
Try to combine some of the smaller paragraphs so the information can flow better.
Materials and Methods -
I am not sure the entire Table 1 needs to be in this section. Maybe a mention that EFA was conducted and the results indicated information that afforded the advance of the research.
Results -
Be sure to place an * indicating the statistically significant findings.
I like the display of information. A lot to take in, but consumable.
Discussion -
Real check the grammar in this section.
Vague terms like 'lots and some' should be elevated.
Link the limitations section into one cohesive paragraph.
Conclusion -
Change the term 'will' to 'may be useful'.
Check grammar.
References -
3. 17 and 18 do not have bolded dates as the other do.
I believe after grammatical issues are sorted out this manuscript will be well worth consideration for publication. I hope you find the suggestions above helpful.
Author Response
Dear reviewer,
We are very grateful for your comments about the manuscript. According to your previous advice, we amended the relevant parts of the paper. All revisions to the document have been clearly highlighted in the paper. After these revisions (your professional comments), the quality of this article has been greatly improved. I attached the specific modifications as a file.
It was revised to improve the overall English quality of the paper.
I revised the paper because there were instructions from other reviewers.
Thank you very much for reviewing your thesis.
Kind regards,

Round 2
Reviewer 1 Report
Authors adressed to my concerns.
Author Response
Dear reviewer,
I revised the paper because there were instructions from other reviewers.
Thank you very much for reviewing your thesis.
Kind regards,

Reviewer 2 Report
The information you have offered is an improvement from the first product I was able to review. This elevation in product should be acknowledged. I believe with a few edits your manuscript will be even more refined. The content of the document is nicely presented and thoroughly researched. Please find the comments/suggestions below as an attempt to assist in the refining process.
Abstract –
L13 – shift ‘showed’ to ‘indicate’
L15 – remove ‘concretely’
L21 – shift ‘they’ to ‘athlete’ or ‘participant’
L23-25 – I am not sure this information is needed in the Abstract.
Introduction –
L30 – excellent clarification – well done!
L31- perhaps: 1960s. As a result of this enactment sports began to be further promoted….
L35 – maybe adding parenthetically the respect sports the referred athletes competed in might add context to the information.
L39-45 – I believe you can collapse these lines to blend L38 into the first sentence of the next paragraph. The listing of the accomplishments (although grand) appears a bit dictation like.
L 50 – shift ‘problems’ to ‘challenges’.
L57 – Rewrite as – Fencing is representative of an unpopular sport in Korea.
L63-66 – I am not sure this text adds substantively to the information being presented.
L72 – add the year of the London Olympics
L80 – please consider the following wording change. ‘If growth in the sport of fencing is to continue in Korea, it is important…..’ (if you use the term, first. You must also follow that with, Second).
L91 – you use PBC without offering the meaning. At the first use of PBC, please articulate that as Perceived Behavioral Control (PBC). – This was done in line 97. Just move the syntax
Materials and Methods –
L151 – change ‘we’ to ‘the researchers’
L159 – remove ‘namely’
Nicely articulated
Results –
L202 – I am not a fan of leading a sentence with initials.
Very organized.
Discussion –
L259 – Remove ‘First’
L270 – the positive thoughts of people around them that were received.
L279 – Break into two sentences. …as well. In turn, this can further…
L281 – Remove ‘Second’
L301 – Remove ‘Third’
L321 – change ‘they’ to ‘Members of this population…’
L341 – Remove ‘Fourth’
L371 – consider changing the wording to ‘In an effort to encapsulate the information presented above, the results of this study suggest…..’
L375 – Shift the language to: The following limitations are recognized:’
Conclusion –
Well done!
References –
I am not sure the bolding of publication dates is mandated by the manuscript guidelines. If it is not, I suggest removing all of the bolded dates to seek greater consistency.
Author Response
Dear reviewer,
We are very grateful for your comments about the manuscript. According to your previous advice, we amended the relevant parts of the paper. All revisions to the document have been clearly highlighted in the paper. After these revisions (your professional comments), the quality of this article has been greatly improved. I attached the specific modifications as a file.
Kind regards,
